# Convolutional Neural Fabrics

**Shreyas Saxena**      **Jakob Verbeek**
INRIA Grenoble – Laboratoire Jean Kuntzmann

## Abstract

Despite the success of CNNs, selecting the optimal architecture for a given task remains an open problem. Instead of aiming to select a single optimal architecture, we propose a "fabric" that embeds an exponentially large number of architectures. The fabric consists of a 3D trellis that connects response maps at different layers, scales, and channels with a sparse homogeneous local connectivity pattern. The only hyper-parameters of a fabric are the number of channels and layers. While individual architectures can be recovered as paths, the fabric can in addition ensemble all embedded architectures together, sharing their weights where their paths overlap. Parameters can be learned using standard methods based on back-propagation, at a cost that scales linearly in the fabric size. We present benchmark results competitive with the state of the art for image classification on MNIST and CIFAR10, and for semantic segmentation on the Part Labels dataset.

## 1 Introduction

Convolutional neural networks (CNNs) [15] have proven extremely successful for a wide range of computer vision problems and other applications. In particular, the results of Krizhevsky *et al*. [13] have caused a major paradigm shift in computer vision from models relying in part on hand-crafted features, to end-to-end trainable systems from the pixels upwards. One of the main problems that holds back further progress using CNNs, as well as deconvolutional variants [24, 26] used for semantic segmentation, is the lack of efficient systematic ways to explore the discrete and exponentially large architecture space. To appreciate the number of possible architectures, consider a standard chain-structured CNN architecture for image classification. The architecture is determined by the following hyper-parameters: (i) number of layers, (ii) number of channels per layer, (iii) filter size per layer, (iv) stride per layer, (v) number of pooling *vs*. convolutional layers, (vi) type of pooling operator per layer, (vii) size of the pooling regions, (viii) ordering of pooling and convolutional layers, (ix) channel connectivity pattern between layers, and (x) type of activation, *e.g*. ReLU or MaxOut, per layer. The number of resulting architectures clearly does not allow for (near) exhaustive exploration.

We show that all network architectures that can be obtained for various choices of the above ten hyper-parameters are embedded in a "fabric" of convolution and pooling operators. Concretely, the fabric is a three-dimensional trellis of response maps of various resolutions, with only local connections across neighboring layers, scales, and channels. See Figure 1 for a schematic illustration of how fabrics embed different architectures. Each activation in a fabric is computed as a linear function followed by a non-linearity from a multi-dimensional neighborhood (spatial/temporal input dimensions, a scale dimension and a channel dimension) in the previous layer. Setting the only two hyper-parameters, number of layers and channels, is not ciritical as long as they are large enough. We also consider two variants, one in which the channels are fully connected instead of sparsely, and another in which the number of channels doubles if we move to a coarser scale. The latter allows for one to two orders of magnitude more channels, while increasing memory requirements by only 50%.

All chain-structured network architectures embedded in the fabric can be recovered by appropriately setting certain connections to zero, so that only a single processing path is active between input and output. General, non-path, weight settings correspond to ensembling many architectures together,

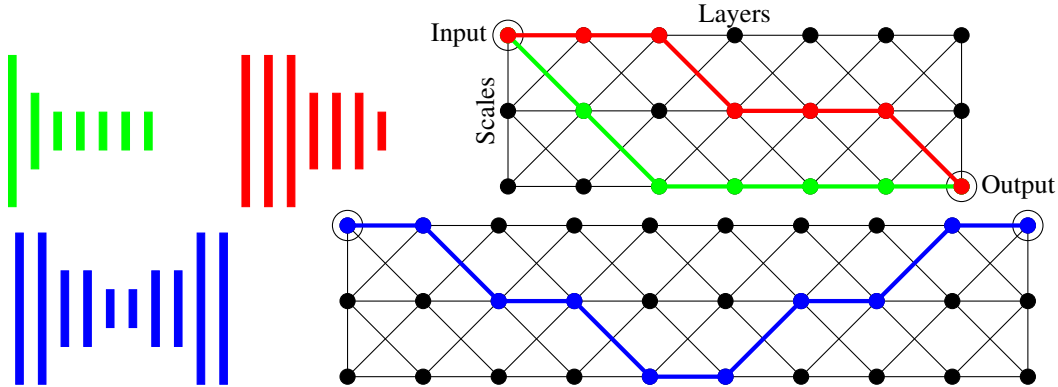

Figure 1: Fabrics embedding two seven-layer CNNs (red, green) and a ten-layer deconvolutional network (blue). Feature map size of the CNN layers are given by height. Fabric nodes receiving input and producing output are encircled. All edges are oriented to the right, down in the first layer, and towards the output in the last layer. The channel dimension of the 3D fabric is omitted for clarity.

which share parameters where the paths overlap. The acyclic trellis structure allows for learning using standard error back-propagation methods. Learning can thus efficiently configure the fabric to implement each one of exponentially many embedded architectures, as well as ensembles of them. Experimental results competitive with the state of the art validate the effectiveness of our approach.

The contributions of our work are: (1) Fabrics allow by and large to sidestep the CNN model architecture selection problem. Avoiding explicitly training and evaluating individual architectures using, *e.g.*, local-search strategies [2]. (2) While scaling linearly in terms of computation and memory requirements, our approach leverages exponentially many chain-structured architectures in parallel by massively sharing weights among them. (3) Since our fabric is multi-scale by construction, it can naturally generate output at multiple resolutions, *e.g.* for image classification and semantic segmentation or multi-scale object detection, within a single non-branching network structure.

## 2   Related work

Several chain-structured CNN architectures, including Alex-net [13] and the VGG-16 and VGG-19 networks [27], are widely used for image classification and related tasks. Although very effective, it is not clear that these architectures are the best ones given their computational and memory requirements. Their widespread adoption is in large part due to the lack of more effective methods to find good architectures than trying them one-by-one, possibly initializing parameters from related ones [2].

CNN architectures for semantic segmentation, as well as other structured prediction tasks such as human pose estimation [25], are often derived from ones developed for image classification, see *e.g.* [20, 24, 31, 33]. Up-sampling operators are used to increase the resolution of the output, compensating for pooling operators used in earlier layers of the network [24]. Ronneberger *et al.* [26] present a network with additional links that couple layers with the same resolution near the input and output. Other architectures, see *e.g.* [3, 7], process the input in parallel across several resolutions, and then fuse all streams by re-sampling to the output resolution. Such architectures induce networks with multiple parallel paths from input to output. We will show that nearly all such networks are embedded in our fabrics, either as paths or other simple sub-graphs.

While multi-dimensional networks have been proposed in the past, *e.g.* to process non-sequential data with recurrent nets [5, 11], to the best of our knowledge they have not been explored as a "basis" to span large classes of convolutional neural networks. Misra *et al.* [23] propose related cross-stitch networks that exchange information across corresponding layers of two copies of the same architecture that produces two different outputs. Their approach is based on Alex-net [13], and does not address the network architecture selection problem. In related work Zhou *et al.* [34] interlink CNNs that take input from re-scaled versions of the input image. The structure of their network is related to our fabric, but lacks a sparse connectivity pattern across channels. They consider their networks for semantic segmentation, and set the filter sizes per node manually, and

use strided max-pooling for down-sampling and nearest neighbor interpolation for up-sampling. The contribution of our work is to show that a similar network structure suffice to span a vast class of network architectures for both dense prediction and classification tasks.

Springenberg *et al*. [29] experimentally observed that the use of max-pooling in CNN architectures is not always beneficial as opposed to using strided convolutions. In our work we go one step further and show that ReLU units and strided convolutions suffice to implement max-pooling operators in our fabrics. Their work, similar to ours, also strives to simplify architecture design. Our results, however, reach much further than only removing pooling operators from the architectural elements. Lee *et al*. [17] generalize the max and average pooling operators by computing both max and average pooling, and then fusing the result in a possibly data-driven manner. Our fabrics also generalize max and average pooling, but instead of adding elementary operators, we show that settings weights in a network with fewer elementary operators is enough for this generalization.

Kulkarni *et al*. [14] use $\ell_1$ regularization to automatically select the number of units in "fully-connected" layers of CNN architectures for classification. Although their approach does not directly extend to determine more general architectural design choices, it might be possible to use such regularization techniques to select the number of channels and/or layers of our fabrics.

Dropout [30] and swapout [28] are stochastic training methods related to our work. They can be understood as approximately averaging over an exponential number of variations of a given architecture. Our approach, on the other hand, allows to leverage an exponentially large class of architectures (ordering of pooling and convolutional layers, type of pooling operator, *etc*.) by means of continuous optimization. Note that these approaches are orthogonal and can be applied to fabrics.

## 3 The fabric of convolutional neural networks

In this section we give a precise definition of convolutional neural fabrics, and show in Section 3.2 that most architectural network design choices become irrelevant for sufficiently large fabrics. Finally, we analyze the number of response maps, parameters, and activations of fabrics in Section 3.3.

### 3.1 Weaving the convolutional neural fabric

Each node in the fabric represents one response map with the same dimension $D$ as the input signal ($D = 1$ for audio, $D = 2$ for images, $D = 3$ for video). The fabric over the nodes is spanned by three axes. A **layer axis** along which all edges advance, which rules out any cycles, and which is analogous to the depth axis of a CNN. A **scale axis** along which response maps of different resolutions are organized from fine to coarse, neighboring resolutions are separated by a factor two. A **channel axis** along which different response maps of the same scale and layer are organized. We use $S = 1 + \log_2 N$ scales when we process inputs of size $N^D$, *e.g.* for 32×32 images we use six scales, so as to obtain a scale pyramid from the full input resolution to the coarsest 1×1 response maps.

We now define a sparse and homogeneous edge structure. Each node is connected to a 3×3 scale–channel neighborhood in the previous layer, *i.e.* activations at channel $c$, scale $s$, and layer $l$ are computed as $a(s, c, l) = \sum_{i,j \in \{-1,0,1\}} \text{conv}(a(c + i, s + j, l - 1), w_{scl}^{ij})$. Input from a finer scale is obtained via strided convolution, and input from a coarser scale by convolution after upsampling by padding zeros around the activations at the coarser level. All convolutions use kernel size 3. Activations are thus a linear function over multi-dimensional neighborhoods, *i.e.* a four dimensional 3×3×3×3 neighborhood when processing 2D images. The propagation is, however, only convolutional across the input dimensions, and not across the scale and layer axes. The "fully connected" layers of a CNN correspond to nodes along the coarsest 1×1 scale of the fabric. Rectified linear units (ReLUs) are used at all nodes. Figure 1 illustrates the connectivity pattern in 2D, omitting the channel dimension for clarity. The supplementary material contains an illustration of the 3D fabric structure.

All channels in the first layer at the input resolution are connected to all channels of the input signal. The first layer contains additional edges to distribute the signal across coarser scales, see the vertical edges in Figure 1. More precisely, within the first layer, channel $c$ at scale $s$ receives input from channels $c + \{-1, 0, 1\}$ from scale $s - 1$. Similarly, edges within the last layer collect the signal towards the output. Note that these additional edges do not create any cycles, and that the edge-structure within the first and last layer is reminiscent of the 2D trellis in Figure 1.

### 3.2 Stitching convolutional neural networks on the fabric

We now demonstrate how various architectural choices can be "implemented" in fabrics, demonstrating they subsume an exponentially large class of network architectures. Learning will configure a fabric to behave as one architecture or another, but more generally as an ensemble of many of them. For all but the last of the following paragraphs, it is sufficient to consider a 2D trellis, as in Figure 1, where each node contains the response maps of $C$ channels with dense connectivity among channels.

**Re-sampling operators.** A variety of re-sampling operators is available in fabrics, here we discuss ones with small receptive fields, larger ones are obtained by repetition. Stride-two convolutions are used in fabrics on fine-to-coarse edges, larger strides are obtained by repetition. *Average pooling* is obtained in fabrics by striding a uniform filter. Coarse-to-fine edges in fabrics up-sample by padding zeros around the coarse activations and then applying convolution. For factor-2 *bilinear interpolation* we use a filter that has $1$ in the center, $1/4$ on corners, and $1/2$ elsewhere. *Nearest neighbor interpolation* is obtained using a filter that is $1$ in the four top-left entries and zero elsewhere.

For *max-pooling* over a $2 \times 2$ region, let $a$ and $b$ represent the values of two vertically neighboring pixels. Use one layer and three channels to compute $(a + b)/2$, $(a - b)/2$, and $(b - a)/2$. After ReLU, a second layer can compute the sum of the three terms, which equals $\max(a, b)$. Each pixel now contains the maximum of its value and that of its vertical neighbor. Repeating the same in the horizontal direction, and sub-sampling by a factor two, gives the output of $2 \times 2$ max-pooling. The same process can also be used to show that a network of *MaxOut units* [4] can be implemented in a network of ReLU units. Although ReLU and MaxOut are thus equivalent in terms of the functions they can implement, for training efficiency it may be more advantageous to use MaxOut networks.

**Filter sizes.** To implement a $5 \times 5$ filter we first compute nine intermediate channels to obtain a vectorized version of the $3 \times 3$ neighborhood at each pixel, using filters that contain a single 1, and are zero elsewhere. A second $3 \times 3$ convolution can then aggregate values across the original $5 \times 5$ patch, and output the desired convolution. Any $5 \times 5$ filter can be implemented exactly in this way, not only approximated by factorization, *c.f.* [27]. Repetition allows to obtain filters of any desired size.

**Ordering convolution and re-sampling.** As shown in Figure 1, chain-structured networks correspond to paths in our fabrics. If weights on edges outside a path are set to zero, a chain-structured network with a particular sequencing of convolutions and re-sampling operators is obtained. A trellis that spans $S + 1$ scales and $L + 1$ layers contains more than $\binom{L}{S}$ chain-structured CNNs, since this corresponds to the number of ways to spread $S$ sub-sampling operators across the $L$ steps to go from the first to the last layer. More CNNs are embedded, *e.g.* by exploiting edges within the first and last layer, or by including intermediate up-sampling operators. Networks beyond chain-structured ones, see *e.g.* [3, 20, 26], are also embedded in the trellis, by activating a larger subset of edges than a single path, *e.g.* a tree structure for the multi-scale net of [3].

**Channel connectivity pattern.** Although most networks in the literature use dense connectivity across channels between successive layers, this is not a necessity. Krizhevsky *et al.* [13], for example, use a network that is partially split across two independent processing streams.

In Figure 2 we demonstrate that a fabric which is sparsely connected along the channel axis, suffices to emulate densely connected convolutional layers. This is achieved by copying channels, convolving them, and then locally aggregating them. Both the copy and sum process are based on local channel interactions and convolutions with filters that are either entirely zero, or identity filters which are all zero except for a single 1 in the center. While more efficient constructions exist to represent the densely connected layer in our trellis, the one presented here is simple to understand and suffices to demonstrate feasibility. Note that in practice learning automatically configures the trellis.

Both the copy and sum process generally require more than one layer to execute. In the copying process, intermediate ReLUs do not affect the result since the copied values themselves are non-negative outputs of ReLUs. In the convolve-and-sum process care has to be taken since one convolution might give negative outputs, even if the sum of convolutions is positive. To handle this correctly, it suffices to shift the activations by subtracting from the bias of every convolution $i$ the minimum possible corresponding output $a_i^{\min}$ (which always exists for a bounded input domain). Using the adjusted bias, the output of the convolution is now guaranteed to be non-negative, and to propagate properly in the copy and sum process. In the last step of summing the convolved channels, we can add back $\sum_i a_i^{\min}$ to shift the activations back to recover the desired sum of convolved channels.

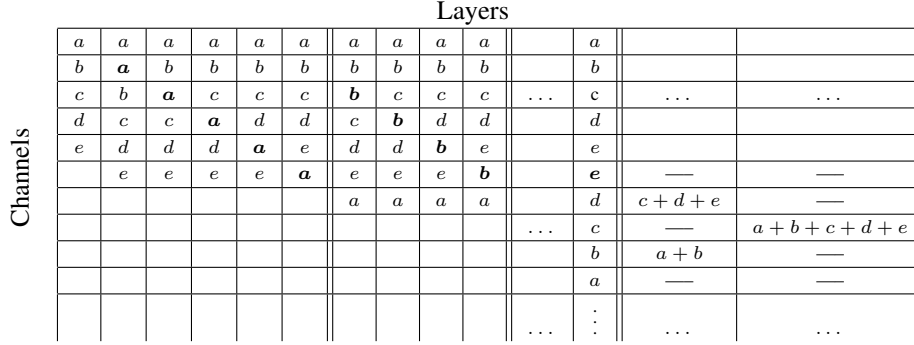

Figure 2: Representation of a dense-channel-connect layer in a fabric with sparse channel connections using copy and swap operations. The five input channels $a, \ldots, e$ are first copied; more copies are generated by repetition. Channels are then convolved and locally aggregated in the last two layers to compute the desired output. Channels in rows, layers in columns, scales are ignored for simplicity.

Table 1: Analysis of fabrics with $L$ layers, $S$ scales, $C$ channels. Number of activations given for $D = 2$ dim. inputs of size $N \times N$ pixels. Channel doubling across scales used in the bottom row.

| # chan. / scale | # resp. maps | # parameters (sparse) | # parameters (dense) | # activations |
|---|---|---|---|---|
| constant | $C \cdot L \cdot S$ | $C \cdot L \cdot 3^{D+1} \cdot 3 \cdot S$ | $C \cdot L \cdot 3^{D+1} \cdot C \cdot S$ | $C \cdot L \cdot N^2 \cdot \frac{4}{3}$ |
| doubling | $C \cdot L \cdot 2^S$ | $C \cdot L \cdot 3^{D+1} \cdot 3 \cdot 2^S$ | $C \cdot L \cdot 3^{D+1} \cdot C \cdot 4^S \cdot \frac{7}{18}$ | $C \cdot L \cdot N^2 \cdot 2$ |

## 3.3 Analysis of the number of parameters and activations

For our analysis we ignore border effects, and consider every node to be an internal one. In the top row of Table 1 we state the total number of response maps throughout the fabric, and the number of parameters when channels are sparsely or densely connected. We also state the number of activations, which determines the memory usage of back-propagation during learning.

While embedding an exponential number of architectures in the number of layers $L$ and channels $C$, the number of activations and thus the memory cost during learning grows only linearly in $C$ and $L$. Since each scale reduces the number of elements by a factor $2^D$, the total number of elements across scales is bounded by $2^D/(2^D - 1)$ times the number of elements $N^D$ at the input resolution.

The number of parameters is linear in the number of layers $L$, and number of scales $S$. For sparsely connected channels, the number of parameters grows also linearly with the number of channels $C$, while it grows quadratically with $C$ in case of dense connectivity.

As an example, the largest models we trained for $32 \times 32$ input have $L = 16$ layers and $C = 256$ channels, resulting in 2M parameters (170M for dense), and 6M activations. For $256 \times 256$ input we used upto $L = 16$ layers and $C = 64$ channels, resulting in 0.7 M parameters (16M for dense), and 89M activations. For reference, the VGG-19 model has 144M parameters and 14M activations.

**Channel-doubling fabrics.** Doubling the number of channels when moving to coarser scales is used in many well-known architectures, see *e.g.* [26, 27]. In the second row of Table 1 we analyze fabrics with channel-doubling instead of a constant number of channels per scale. This results in $C2^S$ channels throughout the scale pyramid in each layer, instead of $CS$ when using a constant number of channels per scale, where we use $C$ to denote the number of "base channels" at the finest resolution. For $32 \times 32$ input images the total number of channels is roughly $11 \times$ larger, while for $256 \times 256$ images we get roughly $57 \times$ more channels. The last column of Table 1 shows that the number of activations, however, grows only by 50% due to the coarsening of the maps.

With dense channel connections and 2D data, the amount of computation per node is constant, as at a coarser resolution there are $4 \times$ less activations, but interactions among $2 \times 2$ more channels. Therefore, in such fabrics the amount of computation grows linearly in the number of scales as compared to a single embedded CNN. For sparse channel connections, we adapt the local connectivity pattern between nodes to accommodate for the varying number channels per scale, see Figure 3 for an illustration. Each node still connects to nine other nodes at the previous layer: two inputs from scale $s - 1$, three from scale $s$, and four from scale $s + 1$. The computational cost thus also grows only

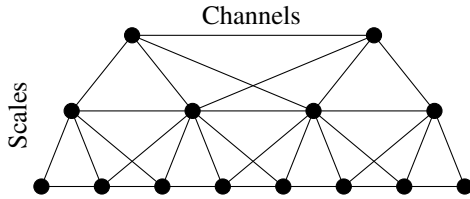

Figure 3: Diagram of sparse channel connectivity from one layer to another in a channel-doubling fabric. Channels are laid out horizontally and scales vertically. Each internal node, *i.e.* response map, is connected to nine nodes at the previous layer: four channels at a coarser resolution, two at a finer resolution, and to itself and neighboring channels at the same resolution.

by 50% as compared to using a constant number of channels per scale. In this case, the number of parameters grows by the same factor $2^S/S$ as the number of channels. In case of dense connections, however, the number of parameters explodes with a factor $\frac{7}{18}4^S/S$. That is, roughly a factor 265 for $32\times32$ input, and 11,327 for $256\times256$ input. Therefore, channel-doubling fabrics appear most useful with sparse channel connectivity. Experiments with channel-doubling fabrics are left for future work.

## 4 Experimental evaluation results

In this section we first present the datasets used in our experiments, followed by evaluation results.

### 4.1 Datasets and experimental protocol

**Part Labels dataset.** This dataset [10] consists of 2,927 face images from the LFW dataset [8], with pixel-level annotations into the classes *hair*, *skin*, and *background*. We use the standard evaluation protocol which specifies training, validation and test sets of 1,500, 500 and 927 images, respectively. We report accuracy at pixel-level and superpixel-level. For superpixel we average the class probabilities over the contained pixels. We used horizontal flipping for data augmentation.

**MNIST.** This dataset [16] consists of $28\times28$ pixel images of the handwritten digits $0,\ldots,9$. We use the standard split of the dataset into 50k training samples, 10k validation samples and 10k test samples. Pixel values are normalized to $[0,1]$ by dividing them by 255. We augment the train data by randomly positioning the original image on a $32\times32$ pixel canvas.

**CIFAR10.** The CIFAR-10 dataset (`http://www.cs.toronto.edu/~kriz/cifar.html`) consists of 50k $32\times32$ training images and 10k testing images in 10 classes. We hold out 5k training images as validation set, and use the remaining 45k as the training set. To augment the data, we follow common practice, see *e.g.* [4, 18], and pad the images with zeros to a $40\times40$ image and then take a random $32\times32$ crop, in addition we add horizontally flipped versions of these images.

**Training.** We train our fabrics using SGD with momentum of 0.9. After each node in the trellis we apply batch normalization [9], and regularize the model with weight decay of $10^{-4}$, but did not apply dropout [30]. We use the validation set to determine the optimal number of training epochs, and then train a final model from the train and validation data and report performance on the test set. We release our Caffe-based implementation at `http://thoth.inrialpes.fr/~verbeek/fabrics`.

### 4.2 Experimental results

For all three datasets we trained sparse and dense fabrics with various numbers of channels and layers. In all cases we used a constant number of channels per scale. The results across all these settings can be found in the supplementary material, here we report only the best results from these. On all three datasets, larger trellises perform comparable or better than smaller ones. So in practice the choice of these only two hyper-parameters of our model is not critical, as long as a large enough trellis is used.

**Part Labels.** On this data set we obtained a super-pixel accuracy of 95.6% using both sparse and dense trellises. In Figure 4 we show two examples of predicted segmentation maps. Table 2 compares our results with the state of the art, both in terms of accuracy and the number of parameters. Our results are slightly worse than [31, 33], but the latter are based on the VGG-16 network. That network has roughly $4,000\times$ more parameters than our sparse trellis, and has been trained from over 1M ImageNet images. We trained our model from scratch using only 2,000 images. Moreover, [10, 19, 31] also include CRF and/or RBM models to encode spatial shape priors. In contrast, our results with convolutional neural fabrics (CNF) are obtained by predicting all pixels independently.

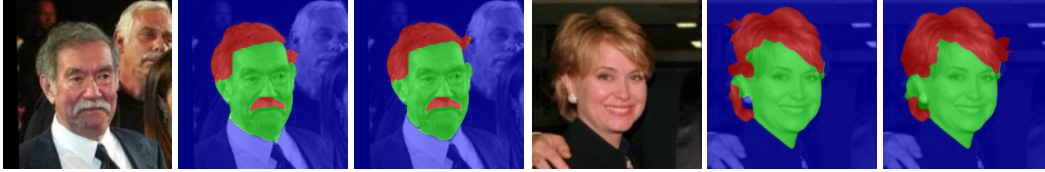

Figure 4: Examples form the Part Labels test set: input image (left), ground-truth labels (middle), and superpixel-level labels from our sparse CNF model with 8 layers and 16 channels (right).

Table 2: Comparison of our results with the state of the art on Part Labels.

|  | Year | # Params. | SP Acccur. | P Accur. |
|---|---|---|---|---|
| Tsogkas *et al*. [31] | 2015 | >414 M | 96.97 | — |
| Zheng *et al*. [33] | 2015 | >138 M | 96.59 | — |
| Liu *et al*. [19] | 2015 | >33 M | — | 95.24 |
| Kae *et al*. [10] | 2013 | 0.7 M | 94.95 | — |
| Ours: CNF-sparse ($L = 8, C = 16$) |  | 0.1 M | 95.58 | 94.60 |
| Ours: CNF-dense ($L = 8, C = 64$) |  | 8.0 M | 95.63 | 94.82 |

**MNIST.** We obtain error rates of 0.48% and 0.33% with sparse and dense fabrics respectively. In Table 3 we compare our results to a selection of recent state-of-the-art work. We excluded several more accurate results reported in the literature, since they are based on significantly more elaborate data augmentation methods. Our result with a densely connected fabric is comparable to those of [32], which use similar data augmentation. Our sparse model, which has $20\times$ less parameters than the dense variant, yields an error of 0.48% which is slightly higher.

**CIFAR10.** In Table 4 we compare our results to the state of the art. Our error rate of 7.43% with a dense fabric is comparable to that reported with MaxOut networks [4]. On this dataset the error of the sparse model, 18.89%, is significantly worse than the dense model. This is either due to a lack of capacity in the sparse model, or due to difficulties in optimization. The best error of 5.84% [22] was obtained using residual connections, without residual connections they report an error of 6.06%.

**Visualization.** In Figure 5 we visualize the connection strengths of learned fabrics with dense channel connectivity. We observe qualitative differences between learned fabrics. The semantic segmentation model (left) immediately distributes the signal across the scale pyramid (first layer/column), and then progressively aggregates the multi-scale signal towards the output. In the CIFAR10 classification model the signal is progressively downsampled, exploiting multiple scales in each layer. The figure shows the result of heuristically pruning (by thresholding) the weakest connections to find a smaller sub-network with good performance. We pruned 67% of the connections while increasing the error only from 7.4% to 8.1% after fine-tuning the fabric with the remaining connections. Notice that all up-sampling connections are deactivated after pruning.

Table 3: Comparison of our results with the state of the art on MNIST. Data augmentation with translation and flipping is denoted by T and F respectively, N denotes no data augmentation.

|  | Year | Augmentation | # Params. | Error (%) |
|---|---|---|---|---|
| Chang *et al*. [1] | 2015 | N | 447K | 0.24 |
| Lee *et al*. [17] | 2015 | N |  | 0.31 |
| Wan *et al*. (Dropconnect) [32] | 2013 | T | 379K | 0.32 |
| CKN [21] | 2014 | N | 43 K | 0.39 |
| Goodfellow *et al*. (MaxOut) [4] | 2013 | N | 420 K | 0.45 |
| Lin *et al*. (Network in Network) [18] | 2013 | N |  | 0.47 |
| Ours: CNF-sparse ($L = 16, C = 32$) |  | T | 249 K | 0.48 |
| Ours: CNF-dense ($L = 8, C = 64$) |  | T | 5.3 M | 0.33 |

Table 4: Comparison of our results with the state of the art on CIFAR10. Data augmentation with translation, flipping, scaling and rotation are denoted by T, F, S and R respectively.

|  | Year | Augmentation | # Params. | Error (%) |
|---|---|---|---|---|
| Mishkin & Matas [22] | 2016 | T+F | 2.5M | 5.84 |
| Lee *et al*. [17] | 2015 | T+F | 1.8M | 6.05 |
| Chang *et al*. [1] | 2015 | T+F | 1.6M | 6.75 |
| Springenberg *et al*. (All Convolutional Net) [29] | 2015 | T+F | 1.3 M | 7.25 |
| Lin *et al*. (Network in Network) [18] | 2013 | T+F | 1 M | 8.81 |
| Wan *et al*. (Dropconnect) [32] | 2013 | T+F+S+R | 19M | 9.32 |
| Goodfellow *et al*. (MaxOut) [4] | 2013 | T+F | >6 M | 9.38 |
| Ours: CNF-sparse ($L = 16, C = 64$) | | T+F | 2M | 18.89 |
| Ours: CNF-dense ($L = 8, C = 128$) | | T+F | 21.2M | 7.43 |

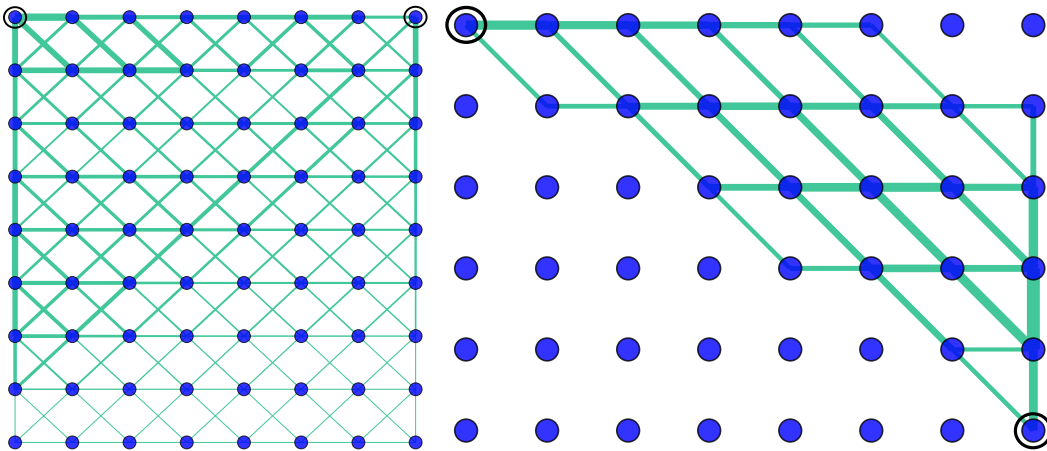

Figure 5: Visualization of mean-squared filter weights in fabrics learned for Part Labels (left) and CIFAR10 (right, pruned network connections). Layers are laid out horizontally, and scales vertically.

## 5 Conclusion

We presented convolutional neural fabrics: homogeneous and locally connected trellises over response maps. Fabrics subsume a large class of convolutional networks. They allow to sidestep the tedious process of specifying, training, and testing individual network architectures in order to find the best ones. While fabrics use more parameters, memory and computation than needed for each of the individual architectures embedded in them, this is far less costly than the resources required to test all embedded architectures one-by-one. Fabrics have only two main hyper-parameters: the number of layers and the number of channels. In practice their setting is not critical: we just need a large enough fabric with enough capacity. We propose variants with dense channel connectivity, and with channel-doubling over scales. The latter strikes a very attractive capacity/memory trade-off.

In our experiments we study performance of fabrics for image classification on MNIST and CIFAR10, and of semantic segmentation on Part Labels. We obtain excellent results that are close to the best reported results in the literature on all three datasets. These results suggest that fabrics are competitive with the best hand-crafted CNN architectures, be it using a larger number of parameters in some cases (but much fewer on Part Labels). We expect that results can be further improved by using better optimization schemes such as Adam [12], using dropout [30] or dropconect [32] regularization, and using MaxOut units [4] or residual units [6] to facilitate training of deep fabrics with many channels.

In ongoing work we experiment with channel-doubling fabrics, and fabrics for joint image classification, object detection, and segmentation. We also explore channel connectivity patterns in between the sparse and dense options used here. Finally, we work on variants that are convolutional along the scale-axis so as to obtain a scale invariant processing that generalizes better across scales.

**Acknowledgment.** We would like to thank NVIDIA for the donation of GPUs used in this research. This work has been partially supported by the LabEx PERSYVAL-Lab (ANR-11-LABX-0025-01).

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
