[Supplementary Material]

# Convolutional Neural Fabrics
# — Supplementary Material —

**Shreyas Saxena**     **Jakob Verbeek**
INRIA Grenoble – Laboratoire Jean Kuntzmann

Figure 1: Visualization of connection strengths in fabrics learned for Part Labels (top left), MNIST (top right), and CIFAR10 (bottom left and right). Layers are laid out horizontally, and scales vertically.

## 1 Fabric visualizations

In Figure 1 we show fabrics learned on the three datasets used in the main paper. The connection strengths are visualized by setting edge widths proportional to the mean squared filter weights related to that connection (mean across the $3 \times 3$ weights and across all channels). The fabrics used for these visualization use dense channel connectivity.

Clearly, the learned connection patterns differ between the segmentation and the classification datasets, since output needs to be produced at different resolutions for these tasks. Interestingly, the pattern is also different between the two classification datasets. For the task of image classification, we have two models, CIFAR10 (bottom-left) and MNIST (top-right). In the case of MNIST, signal in the fabric is propagated in a band-diagonal pattern, exploiting multiple scales in each layer. With CIFAR10 we observe a similar pattern, but with a small difference. The signal is first processed at

the finest scale for a few layers, and then propagated down. These results demonstrate that even for the same task, the fabric is able to configure itself in order to accommodate nuances of the dataset.

For the CIFAR-10 dataset we also show the pruned network, obtained by (arbitrarily) pruning all connections below the mean connection strength. After fine-tuning the pruned network, the classification error is 8.1% as compared to 7.4% for the full fabric, while removing 67% of the connections in the fabric.

## 2   Fabric structure

In Figure 2 we illustrate how the multiple inputs to a node in a fabric are combined. Input from a finer scale is processed by strided convolution, from the same scale by normal convolution, and from a coarser scale by first upsampling the signal by zero padding and then convolving the signal. The three input signals are then added, and then a ReLU activation function is applied to the result.

In Figure 3 we illustrate the 3D fabric structure when using sparse channel connectivity. In this case, each internal node corresponds to a single response map at a given resolution. Nodes receive input from $3 \times 3$ nodes from the scale-channel plane of the previous layer. Note that each edge represents a $3 \times 3$ spatial convolution. Therefore, the input that projects into any "neuron" in the fabric is a 4D tensor of size $3 \times 3 \times 3 \times 3$ across two spatial dimensions, a scale and a channel dimension.

Figure 2: Visualization of the multi-scale signal propagation in fabrics. Layers are laid out horizontally, and scales vertically. Strided convolution and convolution after zero-padding up-sampling are used to adapt finer and coarser input signals to the node resolution.

Figure 3: Visualization of the 3D fabric with sparse channel connectivity. Layers are laid out horizontally, and scales vertically, channels along the third dimension.

Figure 4: Example segmentations on the Part Labels test set, from left to right: input image, ground-truth labels, superpixel-level predictions, and pixel-level prediction from our sparse CNF model with 8 layers and 16 channels. Successful segmentation in the top three rows, failure cases in the bottom three rows. All predictions are made independently across pixels, *i.e.* no CRF models have been used.

# 3 Additional experimental results

In Figure 4 we provide additional segmentation results on the Part Labels data set, both at superpixel and pixel level. In tables 1–8 we give detailed experimental results obtained using sparse and dense fabrics of different sizes, and give the corresponding number of parameters for each model.

The results reported here are measured on the test set, using models trained on the train set only. In the main paper results are reported on the test set using models trained from both the train and validation data. In both cases the number of training epochs has been selected to maximize performance on the validation data, when training from the train data only.

On all three datasets larger fabrics generally give better or comparable results as compared to smaller ones, despite relatively simple regularization by means of weight decay and early stopping. These results show that the number of channels and layers are not critical parameters of our fabrics, it simply suffices to take them large enough.

Table 1: Accuracy on Part Labels for CNF-sparse. Number of parameters given in parentheses.

| Layers / Channels | 4 | | 16 | | 64 | |
|---|---|---|---|---|---|---|
| 2 | 91.95 | *(6K)* | 94.76 | *(23K)* | 95.14 | *(93K)* |
| 4 | 93.94 | *(12K)* | 95.02 | *(47K)* | 95.34 | *(187K)* |
| 8 | 94.87 | *(23K)* | **95.48** | *(93K)* | 95.46 | *(373K)* |
| 16 | 95.15 | *(47K)* | 95.38 | *(187K)* | 95.26 | *(746K)* |

Table 2: Accuracy on Part Labels for CNF-dense. Number of parameters given in parentheses.

| Layers / Channels | 4 | | 16 | | 64 | |
|---|---|---|---|---|---|---|
| 2 | 93.57 | *(8K)* | 95.26 | *(124K)* | 95.33 | *(2M)* |
| 4 | 93.67 | *(16K)* | 95.05 | *(249K)* | 95.20 | *(4M)* |
| 8 | 95.09 | *(31K)* | 95.22 | *(498K)* | **95.39** | *(8M)* |
| 16 | 94.92 | *(62K)* | 95.29 | *(995K)* | 95.34 | *(16M)* |

Table 3: Error rate on MNIST for CNF-sparse. Number of parameters given in parentheses.

| Layers / Channels | 4 | | 8 | | 16 | | 32 | | 64 | | 128 | |
|---|---|---|---|---|---|---|---|---|---|---|---|---|
| 2 | 4.94 | *(4K)* | 2.32 | *(8K)* | 1.68 | *(16K)* | 1.40 | *(31K)* | 0.97 | *(62K)* | 1.00 | *(124K)* |
| 4 | 2.96 | *(8K)* | 1.69 | *(16K)* | 1.14 | *(31K)* | 0.96 | *(62K)* | 0.98 | *(124K)* | 0.78 | *(249K)* |
| 8 | 1.94 | *(16K)* | 1.12 | *(31K)* | 0.87 | *(62K)* | 0.69 | *(124K)* | 0.79 | *(249K)* | 0.57 | *(498K)* |
| 16 | 1.13 | *(31K)* | 0.91 | *(62K)* | 0.71 | *(124K)* | **0.56** | *(249K)* | 0.68 | *(498K)* | 0.70 | *(1M)* |
| 32 | 1.37 | *(62K)* | 0.88 | *(124K)* | 0.67 | *(249K)* | 0.61 | *(498K)* | 0.77 | *(1M)* | 0.73 | *(2M)* |

Table 4: Error rate on MNIST for CNF-dense. Number of parameters given in parentheses.

| Layers / Channels | 4 | | 8 | | 16 | | 32 | | 64 | | 128 | |
|---|---|---|---|---|---|---|---|---|---|---|---|---|
| 2 | 3.47 | *(5K)* | 1.46 | *(21K)* | 0.73 | *(83K)* | 0.65 | *(331K)* | 0.59 | *(1M)* | 0.59 | *(5M)* |
| 4 | 2.74 | *(10K)* | 0.88 | *(42K)* | 0.67 | *(166K)* | 0.54 | *(663K)* | 0.59 | *(3M)* | 0.51 | *(10M)* |
| 8 | 1.65 | *(21K)* | 0.70 | *(83K)* | 0.60 | *(332K)* | 0.52 | *(1M)* | **0.39** | *(5M)* | 0.46 | *(21M)* |
| 16 | 1.04 | *(41K)* | 0.60 | *(166K)* | 0.50 | *(663K)* | 0.55 | *(2M)* | 0.39 | *(10M)* | 0.47 | *(42M)* |
| 32 | 1.29 | *(83K)* | 0.92 | *(332K)* | 0.64 | *(1M)* | 0.57 | *(5M)* | 0.61 | *(21M)* | 0.56 | *(85M)* |

Table 5: Error rate on CIFAR10 for CNF-sparse. Number of parameters given in Table 7.

| Layers / Channels | 2 | 4 | 8 | 16 | 32 | 64 | 128 | 256 |
|---|---|---|---|---|---|---|---|---|
| 2 | 68.70 | 49.65 | 48.95 | 34.48 | 31.48 | 28.82 | 27.67 | 25.56 |
| 4 | 62.34 | 43.69 | 34.28 | 30.07 | 26.18 | 25.14 | 22.96 | 22.60 |
| 8 | 58.26 | 40.02 | 28.10 | 24.44 | 22.12 | 22.20 | 20.66 | 21.38 |
| 16 | 50.31 | 32.28 | 25.70 | 22.65 | 19.74 | 19.07 | 19.05 | **18.89** |

Table 6: Error rate on CIFAR10 for CNF-dense. Number of parameters given in Table 8.

| Layers / Channels | 2 | 4 | 8 | 16 | 32 | 64 | 128 | 256 |
|---|---|---|---|---|---|---|---|---|
| 2 | 68.70 | 50.96 | 33.66 | 23.92 | 18.83 | 15.72 | 13.79 | 13.11 |
| 4 | 62.34 | 41.92 | 27.76 | 19.22 | 14.65 | 13.38 | 12.09 | 10.06 |
| 8 | 58.26 | 35.12 | 22.53 | 15.57 | 13.05 | 10.88 | 9.42 | **9.31** |
| 16 | 50.31 | 28.27 | 19.03 | 13.57 | 10.95 | 9.65 | 10.63 | 14.27 |

Table 7: Number of parameters for CIFAR10, CNF-sparse.

| Layers / Channels | 2 | 4 | 8 | 16 | 32 | 64 | 128 | 256 |
|---|---|---|---|---|---|---|---|---|
| 2 | *(1K)* | *(4K)* | *(8K)* | *(16K)* | *(31K)* | *(62K)* | *(124K)* | *(249K)* |
| 4 | *(3K)* | *(8K)* | *(16K)* | *(31K)* | *(62K)* | *(124K)* | *(249K)* | *(498K)* |
| 8 | *(5K)* | *(16K)* | *(31K)* | *(62K)* | *(124K)* | *(249K)* | *(498K)* | *(1M)* |
| 16 | *(10K)* | *(31K)* | *(62K)* | *(124K)* | *(249K)* | *(498K)* | *(1M)* | *(2M)* |

Table 8: Number of parameters for CIFAR10, CNF-dense.

| Layers / Channels | 2 | 4 | 8 | 16 | 32 | 64 | 128 | 256 |
|---|---|---|---|---|---|---|---|---|
| 2 | *(1K)* | *(5K)* | *(21K)* | *(83K)* | *(331K)* | *(1M)* | *(5M)* | *(21M)* |
| 4 | *(3K)* | *(10K)* | *(42K)* | *(166K)* | *(663K)* | *(3M)* | *(10M)* | *(42M)* |
| 8 | *(5K)* | *(21K)* | *(83K)* | *(332K)* | *(1M)* | *(5M)* | *(21M)* | *(85M)* |
| 16 | *(10K)* | *(41K)* | *(166K)* | *(663K)* | *(2M)* | *(10M)* | *(42M)* | *(170M)* |