[Reviews · NeurIPS 2016]

Reviewer 1

Summary

This paper addresses the problem of how to select certain discrete hyperparameters for deep feedforward convolutional and deconvolutional networks, namely the choice of /when/ required downsampling and upsampling operations take place in the computation path. The solution they offer seems initially quite elegant -- one can construct a "trellis" that ensembles together all the possible architectures in this hyperparameter space (subject to each architecture having the same number of layers, and comparable channel structure), and enforce a kind of weight sharing within the trellis. This should allow one to sidestep the problem of committing to a particular hyperparameter.

Qualitative Assessment

Elegant as the idea sounds, the results are somewhat underwhelming. The naive implementation of this (what the authors call the "dense" model) underperforms while seeming to use considerably more parameters than nets in the literature (Tables 2-4). The authors are careful to quantify the parameter growth, but having 10-20x the parameters of its competitors for cifar10 with worse scores suggests that it's overfitting. The "sparse" channel connectivity model (which wasn't well explained) cuts down the number of parameters to something reasonable, but it appears too constrained to be able to learn a task like cifar10. That's not a good sign for its broader applicability. So while the authors claim that this could be a solution to choosing some hyperparameters, it really does raise a host of other problems. And if convolutional fabrics were the solution, then one is now saddled with the choice of which channel connectivity pattern to use. Also, in the big scheme of things, should we be worried about these particular architectural hyperparameters over others? The authors conclude by saying they still want to try adam/dropout/dropconnect/maxout/residual methods -- that's at least 5 more dimensions to the hyperparameter space that the trellis can't ensemble over. Ultimately, I get the sense that this could be an interesting idea, but it needs to be pushed in a different direction.

Confidence in this Review

2-Confident (read it all; understood it all reasonably well)


Reviewer 2

Summary

This paper tries to eliminate a bunch of architectural hyperparameters in convolutional neural networks by instantiating a trellis of connected layers, which contains many possible architectures as individual paths. The assumption is that the learning procedure will tease out which paths are sensible, rather than the practitioner having to determine the hyperparameters by iterative experiments. The resulting models are evaluated on three datasets (part labels, MNIST, CIFAR-10).

Qualitative Assessment

The idea is quite interesting and timely: eliminating some of the many hyperparameters that need to be tuned in CNN design is a welcome development with potentially high impact. I like how Figure 5 demonstrates that the learning process is indeed capable of configuring the trellis as needed. It is somewhat unfortunate that all experiments in the paper are conducted with fabrics with a constant number of channels per scale: this goes against common practice in designing CNNs and leads to wasted capacity. It means that the size of the representation shrinks as the level of abstraction increases, which is typically counteracted by having more channels at higher abstraction levels. The paper states that experiments with channel doubling are ongoing, but these should really be part of the paper as they are much more relevant. The selection of datasets is also somewhat disappointing: all of them are quite small, and MNIST and CIFAR-10 are not really suitable for benchmarking by themselves anymore. Overall, it seems that the most interesting results (channel-doubling, PASCAL VOC, MS COCO, ImageNet) are not ready yet. The paper would be much more impactful if they were included. (The channel-doubling architecture is also referred to in the conclusion, which I think is inappropriate if no actual experiments with it are described in the paper.) In the conclusion it is claimed that the results are competitive with the state of the art for all datasets, but at least for CIFAR-10 this is not true, even when only taking into account models trained with the same level of data augmentation (translation / flipping only -- an error rate of under 6% has been reported by e.g. Mishkin & Matas, 2016). The performance is very good of course, but the fact that the model needs 32M parameters to achieve it (an order of magnitude more than is typical for this dataset) should also be taken into account. The sparse version is interesting in this respect, but it does not seem to work very well. Perhaps other sparsity patterns (learnt in some way?) could provide a way to instantiate fabrics with a more reasonable number of parameters. The last paragraph of section 3.2 explains how the sparse fabric can implement densely connected convolutional layers, but I am skeptical that the learning procedure could find such a solution with current optimization methods. It would be interesting to show this (but I appreciate that this is non-trivial). Related work should probably also include "Recombinator Networks" by Honari et al. (2015), as the architecture introduced in this paper is quite similar, at least superficially.

Confidence in this Review

3-Expert (read the paper in detail, know the area, quite certain of my opinion)


Reviewer 3

Summary

This paper proposes an interesting approach to address the problem of deciding what architecture to use in a convolutional network. This is done via an elegant construction of a “neural fabric” or trellis which can efficiently subsume exponentially many (related) convolutional nets within it. An individual convnet is realized as a path through the trellis, and the trellis as a who can be seen as being the ensemble of all possible networks embedded therein. The framework is applied to a part-labelling dataset, as well as MNIST and CIFAR 10. The model shows strong results with few parameters on the part labelling task, and also gives competitive results on MNIST and CIFAR10 — albeit at the cost of a relatively large number of parameters to get the best results (using dense channel connectivity).

Qualitative Assessment

The analysis in section 3 usefully demonstrates the potential generality of a sparse trellis, by showing how it can implement denser operations, max-pooling, etc. These are very nice features. However, one concern is although such operations *could* be learned, there is a question of how easy it is to learn them. Somewhat relatedly, in the experiments section the best results on CIFAR and MNIST are achieved using a rather large number of total parameters. The trellis visualizations (Figure 5) seem like a nice way of quickly gaining intuition for structures that might be useful in addressing a particular problem. (One could imagine subsequently performing a focussed search of more elaborate architectures based on those insights.) I found the trellis construction to be novel, and an interesting approach to dealing with the difficulty of finding good architectures. I suspect there might be additional interesting extension to consider — for instance using smaller fabrics as “blocks” or modules within other networks. If the methods here can be shown to scale well to larger problems (e.g. Imagenet, etc) then I would expect this work to have very useful impact in the community. Even with the smaller scale results presented here, I think that there are valuable ideas and insights to be presented.

Confidence in this Review

3-Expert (read the paper in detail, know the area, quite certain of my opinion)


Reviewer 4

Summary

This paper presents an approach, neural fabrics, that tries to learn models that are adaptive to the design choice for layers, scales, and channels. The particular aspect is about trellis that connects response maps. Experimental results on part label, MNIST, and Cifar-10.

Qualitative Assessment

The effort in building a method that automatically explores the architecture design of CNN should be encouraged. This paper presents interesting 3D trellis but the hyper-parameters for number of channels and layers are still left for manual specification. Although the method is interesting, there is no clear advantages of the proposed method, as the results on part label, MNIST, and Cifar-10 are not competitive. The overall model parameter complexity is also very high, making it impractical. The additional overhead in terms of the training complexity is not detailed but might be very high.

Confidence in this Review

2-Confident (read it all; understood it all reasonably well)


Reviewer 5

Summary

The paper introduces a solution to the CNN architecture selection problem. `Neural fabric` term is introduced, which is a 3D trellis that connects response maps at different layers, scales and channels. It is claimed that the trellis hyper-parameters (#chn. and layers) are not performance critical. An interesting property of the trellis is that, it does not only use a particular architecture that emerges, but it may also naturally end up use a weighted combination of different architectures.

Qualitative Assessment

Results in Table 2 and Table 3 show that this is a viable alternative to CNN architecture selection. I am not sure if the improvement warrants an oral paper, but definitely interesting to make a poster and stir discussions about the area.

Confidence in this Review

1-Less confident (might not have understood significant parts)


Reviewer 6

Summary

The authors propose a convolutional neural fabric model, which addresses the issue of selecting an optimal CNN architecture. The proposed "fabric" model consists of 3D trellis, which are essentially the response maps that capture scale, layer, and channel information of the traditional CNN architectures. Each node in the trellis is connected to a 3x3 scale-channel neighborhood from the previous layer. The authors show that the traditional CNN architectures can be embedded and implemented via neural fabric models. Finally, the authors present results on MNIST, CIFAR and "Part Labels" dataset and show competitive results.

Qualitative Assessment

I think the authors present an interesting and a novel idea that addresses an important problem of how to select an optimal CNN architecture. I think the major strength of this paper is the novelty in the technical approach. I had several comments in regards to the experiments. As is standard for many deep learning architecture papers, the authors conduct experiments on MNIST dataset, CIFAR, and in this case "Part Labels" dataset. In my opinion, the tasks related to these datasets (especially MNIST) are not very challenging, and the results are extremely saturated (over >95% accuracy). I would like to see some of the experiments on more challenging tasks such as semantic segmentation on PASCAL, etc. Having said this, I think the key contribution of this paper is the novelty of the method, and I would be willing to overlook the experiments on these "less challenging" tasks, especially since it seems like this is a common practice in this field (a practice, which is not good in my opinion). Additionally, I also didn't find some of the experiments that convincing. I thought that the results for the Parts Labeling task were solid. However, the disparities between the error rates on the MNIST dataset seem pretty large (considering how saturated the performance is). Especially if we look at the top 2 baseline methods, we observe that without using data augmentation , these methods achieve better results with significantly less number of parameters (>3M) than CNF-dense. The same trend occurs on the CIFAR dataset, where we have top 2 methods significantly outperform CNF-dense method with almost 30M less parameters than CNF-dense. Therefore, based on these experiments, I am not convinced that the method would work very well on some of the more challenging tasks (such as semantic segmentation). In summary, even though I didn't find many of these experiments convincing, I would still like to see this paper published. There has been a lot of incremental work on CNNs recently, that do not contribute much to the field despite very good performance in certain tasks. In comparison, even though this paper may not have the most impressive empirical results, it presents a novel alternative to CNNs, which addresses an important CNN architecture selection problem. ---POST REBUTTAL--- I read the rebuttal. I agree with AR5 that the authors should include more experiments clearly outlining theoretical and practical costs of training compared to the traditional CNNs. As I mentioned previously, I am not convinced that the proposed approach is practical at its current state. It doesn't seem to outperform other techniques on the more challenging datasets, even if it uses a larger number of parameters than other methods. However, I like the idea of the paper, and I think that it could have a solid impact on deep learning community, as the CNN architecture selection problem is an important one.

Confidence in this Review

2-Confident (read it all; understood it all reasonably well)